# All Fiber Vector Magnetometer Based on Nitrogen-Vacancy Center

**DOI:** 10.3390/nano13050949

**Published:** 2023-03-06

**Authors:** Man Zhao, Qijing Lin, Qingzhi Meng, Wenjun Shan, Liangquan Zhu, Yao Chen, Tao Liu, Libo Zhao, Zhuangde Jiang

**Affiliations:** 1State Key Laboratory of Mechanical Manufacturing Systems Engineering, Xi’an Jiaotong University, Xi’an 710049, China; 2Collaborative Innovation Center of High-End Manufacturing Equipment, Xi’an Jiaotong University, Xi’an 710054, China; 3Shandong Laboratory of Yantai Advanced Materials and Green Manufacturing, Yantai 265503, China; 4Xi’an Jiaotong University (Yantai) Research Institute for Intelligent Sensing Technology and System, Xi’an Jiaotong University, Xi’an 710049, China

**Keywords:** magnetometer, nitrogen-vacancy centers, fiber, vector magnetic field

## Abstract

Magnetometers based on nitrogen-vacancy (NV) centers in diamonds have promising applications in fields of living systems biology, condensed matter physics, and industry. This paper proposes a portable and flexible all-fiber NV center vector magnetometer by using fibers to substitute all conventional spatial optical elements, realizing laser excitation and fluorescence collection of micro-diamond with multi-mode fibers simultaneously and efficiently. An optical model is established to investigate multi-mode fiber interrogation of micro-diamond to estimate the optical performance of NV center system. A new analysis method is proposed to extract the magnitude and direction of the magnetic field, combining the morphology of the micro-diamond, thus realizing μm-scale vector magnetic field detection at the tip of the fiber probe. Experimental testing shows our fabricated magnetometer has a sensitivity of 0.73 nT/Hz^1/2^, demonstrating its feasibility and performance in comparison with conventional confocal NV center magnetometers. This research presents a robust and compact magnetic endoscopy and remote-magnetic measurement approach, which will substantially promote the practical application of magnetometers based on NV centers.

## 1. Introduction

The nitrogen-vacancy (NV) center consists of a substitutional nitrogen atom (N) combined with a vacancy (V) in one of the nearest neighbor sites of the diamond crystal lattice. Due to the excellent properties of long-lived spin coherence and fluorescence stability, nitrogen–vacancy centers in diamond offer an unprecedented solid-state platform for magnetometers and thermometers with hypersensitivity and ultra-high spatial resolution as an atomic-scale defect in the diamond crystal [1,2,3,4,5,6]. NV center magnetometers have the potential of miniaturization and practicality at room temperature under Earth’s magnetic field background. Other state-of-the-art magnetometers, such as superconducting quantum interference devices and spin-exchange relaxation-free magnetometers, exhibit sensitivities at the fT/Hz^1/2^ level, which is a thousand times higher than NV center magnetometer’s sensitivity. However, they require more rigorous working conditions and longer measurement times, which greatly limit their application in practical engineering. In contrast, NV center magnetometers have the characteristics of fast start-up and can work in complex physical field environments such as high temperature and high pressure. NV center magnetometers have been widely used in room-temperature measurement of single living cells [7], high resolution imaging of neural activity [8], as well as magnetic field imaging of advanced material [9,10] in recent years.

However, the most excellent results of NV center magnetometers were achieved in a lab environment with a bulk device. It is necessary to develop compact, integrated, and mobile NV sensors for industrial applications. Fiber integration is an essential method to enable NV center sensors for practical application. Fiber-based NV center sensors have been providing compact and practical solid-state tools for bio-imaging, chip inspection, and nondestructive testing [11,12,13,14,15]. Some studies have been conducted by attaching diamonds varying in size from tens to hundreds of microns at the end of the multi-mode fiber, taper fiber, and photonic crystal fiber for highly efficient laser excitation and fluorescence collection, with magnetometer sensitivity in the ranges of 1.5–300 nT/Hz^1/2^ [16,17,18,19,20]. Fiber taper and photonic crystal fiber were applied to enhance fluorescence collection efficiency because of their large numerical aperture. These fiber-based NV center sensors have also been expanded to magnetometer scanning probe or remote measuring probe. However, there still remain several space light optical elements that need to be precisely spaced, which makes the system unstable and inflexible. Moreover, the space light path is easily interfered with in practical use. Space optical elements, such as objective and dichroic lenses, greatly limit the practical application of the NV center magnetometer. In previous studies, a 1 × 2 fiber optical coupler was applied instead of a dichroic mirror to separate laser and fluorescence [20,21], but its relatively low optical transmission efficiency restricted the sensor’s performance.

To address the aforementioned issues, we present an all-fiber NV center system based on multi-fiber optical elements with high fluorescence efficiency and extraordinary optical transmission efficiency. In addition, the all-fiber NV center magnetometer can be quickly set up outside the lab and operate in complex environments, which presented a robust and compact magnetic endoscopy and remote magnetic measurement approach. Due to the small size of the diamond particles, the all-fiber NV center magnetometer could also realize μm-scale magnetic field detection. Optical interrogation of the micro-diamond in the confocal system and fiber system was investigated to estimate the optical performance of the NV center system. The all-fiber NV center magnetometer was achieved with a sensitivity of 0.73 nT/Hz^1/2^, presenting a robust and compact magnetic endoscopy and remote magnetic measurement approach. 

In some published works by other groups, a three-axes Helmholtz coil and repetitive optically detected magnetic resonance (ODMR) scans with magnetic field rotation were used to identify the relative orientation of NV center axes in the lab-frame coordinate and measure the direction and amplitude of the vector magnetic field [22,23]. This method was complicated and time-consuming. Based on the all-fiber NV magnetometers, we also proposed a simplified and rapid approach for vector magnetic field measurement, considering the relationship between the morphology of micro-diamond and NV centers orientations [24]. Using a static magnetic field with a known direction ***B_a_*** provided by one axis coil, we could determine the relative orientation of the NV center axes and establish the transformation relationship between lab-based Cartesian system and NV center axis coordinate system with a single ODMR scan. Following that, ODMR scans of the measured magnetic field ***B_m_*** and superimposed magnetic field ***B_a_*** +***B_m_*** are conducted to measure the amplitude and direction of ***B_m_*** in lab-based Cartesian system.

## 2. Materials and Methods

The fiber-based magnetometer is shown in Figure 1a. The 532 nm laser (100 mW) is first sent to a fiber-connected acousto-optic modulator (AOM, S-M200-0.4C2C-3-F2S, Gooch & Housego, Ilminster Somerset, UK) to achieve a quick-switch laser pulse for pulsed ODMR measurement. Compared with continuous ODMR measurement, pulsed ODMR technique avoids optical and MW power broadening of the spin resonances, which can achieve higher sensitivity. Then the laser is guided to the wideband multi-mode circulator (WMC, WMC3L1F, Thorlabs, NJ, USA) from Port 1. About 98% of the laser power from Port 1 is transmitted to pump the diamond, which is glued at Port 2, as shown in Figure 1b. About 92% of the fluorescence collected by Port 2 (multi-mode fiber, a core = 200 μm, NA_1_ = 0.39) largely spreads to Port 3. The light transmitted from Port 1 to 3 and Port 2 to 1 is isolated. The laser and fluorescence emitted from the diamond are effectively separated by the WMC, replacing the traditional dichroic mirror. The green and red arrows in Figure 1a show the path and direction of the laser and fluorescence, respectively. At the same time, about ten times higher optical transmission efficiency was achieved compared to 90:10, 1 × 2 fiber optical coupler. The fiber collimator, long-wavelength filter, and focusing lens were fixed to an avalanche photodetector (APD, 430A/M, Thorlabs, NJ, USA) through lens tubes and SM1 screw thread to collimate, filter, and focus fluorescence from Port 3 onto the detector’s active area. 

To manipulate the electron spin state of NV centers, a radio-frequency signal was generated by a signal generator (N5171B, Keysight, CA, USA). It was then routed through a microwave switch (ZFSWA2R-63DR-S+, Mini-circuits, NY, USA), an amplifier (ZHL-16W43-S+, Mini-circuits, NY, USA), and finally delivered to the copper coil antenna (Figure 1b). The oscilloscope performed the data acquisition of the detected signal from the APD. Both the AOM and the microwave switch were both controlled by a Programmable TTL Pulse Generator (PG, PulseBlasterESR-500 MHz, Spincore, FL, USA), which is mounted on the computer’s motherboard. To identify the relative orientation of the NV center axes and measure the vector magnetic field, a one-axis Helmholtz coil was used to generate a biased magnetic field.

The micro-diamond we employed was a type-Ib high-pressure, high-temperature (HPHT) diamond with a diameter of about 550 μm, as shown in Figure 1c. The diamond was irradiated with an energy of 10 MeV and a dose of 1 × 10^18^ cm^−2^ electron beam, then annealed at 850 °C in vacuum and 500 °C in the air, yielding a density of NV centers approximately 1.08 × 10^18^ cm^−3^. The micro-diamond was a cubo-octahedron with surfaces of 8 hexagons and 6 quadrilaterals in morphology. The hexagon surfaces correspond to the <111> crystal plane of the diamond. It is effortless to ensure thaqt the <111> crystal orientation of the diamond coincides with the axial direction of the optical fiber by gluing a hexagon surface to the fiber end face. The micro-diamond was deposited on the tip of Port 2 under an optical microscope by ethyl cyanoacrylate glue. To improve the mechanical strength of the fiber probe, the fiber end near the micro diamond was encapsulated by a ceramic ferrule. The diamond was exposed out of the ceramic ferrule, and the copper coil antenna was wound around the ceramic ferrule near the diamond.

## 3. Experiments and Discussion

### 3.1. Optical Interrogation of NV Centers in Diamond

The all-fiber NV center magnetometer was analyzed in terms of optical properties and sensitivity performance, compared with the traditional confocal system. Laser loss in the fiber system mainly occurred at AOM, fiber mating sleeves(FMS), and WMC. The diffraction efficiency of fiber-coupled AOM was 70%. Multi-mode fiber connectors provided low insertion loss, typically less than 0.5 dB, and the insertion loss from single-mode to multi-mode was negligible. The laser power transmitted to the diamond reached 64.2 mW when the laser source power was set to 100 mW, as measured by an optical power meter. In addition, the laser transmission efficiency of the traditional space light confocal system was about 62.3%, as shown in Figure 2a. As a result, we could conclude that the two systems above had similar laser transmission efficiency. The fluorescence intensity *I_0_* as a function of laser source power was measured in the confocal system (NA_2_ = 0.9, objective) and the fiber system. As Figure 2b illustrated, the fluorescence intensity detected in the fiber system was about 1.87 times higher than that in the confocal system with the same laser source power. The sensitivity of the quantum magnetometer was inversely proportional to *I_0_*^1/2^ [25]. Therefore, the sensitivity performance of the fiber optical system was 27% higher than that of the confocal system.

As shown in Figure 3a, the diamond region (the red area) illuminated by the laser is a circular truncated cone. The diameter of the upper surface is *a_0_* ≈ *a_core_* = 200 μm, and the divergence angle and acceptance angle are both *θ_d_ =* arcsin(NA_1_/*n_d_*) = 9.23°, where refractive index of diamond *n_d_* = 2.43 and the numerical aperture of Port2 NA_1_ = 0.39. The collecting efficiency of the optical fiber is estimated using geometric optics and the Fresnel reflection law [26]. The collecting efficiency of the fiber system is given by Equation (1). NV centers in micro-diamonds could be treated as a uniformly radiating point light source. The emitted fluorescence intensity distribution (*I_p_*) of NV centers is 1/4π.
(1)$$\beta_{f}=\int_{0}^{\theta_{d}}\int_{0}^{2\pi }I_{p}\frac{4n_{core}n_{d}}{{\left(n_{core}+n_{d}\right)}^{2}}\sin\theta d\varphi d\theta$$

The collecting efficiency of the fiber system is estimated as 0.607% with a refractive index of the fiber core *n_core_* = 1.459. For the laser-excited region in the diamond, the laser power through the axial cross-section is consistent. In each tiny cylinder of the excited region with a thickness of *d*z, the power of the emitted fluorescence is (*λ_laser_/λ_fluo_)δηN*_0_*P_laser_*, where *λ_laser_* = 532 nm and the fluorescence wavelength range of NV center is 600–800 nm (Figure 3b). Here we used *λ_fluo_* = 700 nm to estimate the fluorescence intensity. *δ* ≈ 3 × 10^−17^ cm^2^ is the absorption cross section of type Ib diamond, *η* ≈ 0.1 is the PL quantum yield [27], *N*_0_ = 1.08 × 10^18^/cm^3^ is the NV center density, and *P_laser_* = 64.2 mW is the laser power exciting the diamond. The fluorescence power collected by the fiber is written as:(2)$$P_{f}=\frac{\lambda_{laser}}{\lambda_{fluo}}\delta \eta N_{0}P_{\mathrm{laser}}\beta_{f}z_{d}$$
where *z_d_* ≈ 550 μm is the micro-diamond dimension in the axial direction of the optical fiber. *P_f_* = 5 × 10^−2^ mW is obtained by calculation with Equation (2). The optical loss from Port 2 of the WDM to the active area of the APD results in a transmission efficiency of about 87%. As a result, the fluorescence power detected by the APD is theoretically 4.35 × 10^−2^ mW. 

Experimentally, when the transimpedance gain was tuned to 2.15 × 10^4^ *v*/*w* (700 nm), the APD detected fluorescence with an output voltage of 885.6 mV, which equates to 4.12 × 10^−2^ mW. There is an acceptable deviation of 5.6% between the APD detected value and the theoretically estimated value. The following factors contribute to this difference. To begin with, there is a minor discrepancy between these parameters in Equation (2) and their actual values. Secondly, at the radial marginal region of the excited region, the fluorescence collecting efficiency is generally lower than 0.607%.

In the confocal system, the excited area of diamond is the green part, as shown in Figure 4a. The collecting efficiency of the confocal system is given by Equation (3). Since the multi-mode fiber with a 200 μm fiber core serves as the pinhole, only the fluorescence radiating near the focus is collected and transmitted to the detector. The fluorescence acceptance angle is *θ*_0_
*=* arcsin(NA_2_/*n_d_*) = 21.74°.
(3)$$\beta_{c}=\int_{0}^{\theta_{0}}\int_{0}^{2\pi }I_{p}\frac{4n_{\mathrm{air}}n_{d}}{{\left(n_{\mathrm{air}}+n_{d}\right)}^{2}}\sin\theta d\varphi d\theta ,$$

The collecting efficiency of the confocal system is estimated as 2.937% with a refractive index of the fiber core *n_air_* = 1. An electronically controlled translation table controls the axial movement of a plane mirror to measure the axial optical sectioning curve of the confocal system, plotted in Figure 4b. The normalized axial optical sectioning curve indicates the axial optical transmission efficiency of fluorescence for every axial cross-section because defocus fluorescence signal is filtered by the pinhole [28]. The full width at half-maximum(*z_c_*) is used to estimate the thickness at which diamond fluorescence is detected: *z_c_* = 62 μm. The fluorescence power collected by the objective is written as:(4)$$P_{c}=\frac{\lambda_{laser}}{\lambda_{fluo}}\delta \eta N_{0}P_{laser}\beta_{c}z_{c},$$

According to Equations (2) and (4), we further deduced that:$$\frac{P_{f}}{P_{c}}=\frac{\beta_{f}z_{d}}{\beta_{c}z_{c}}=\frac{0.607\%\times 560}{2.937\%\times 62}=1.846\approx \frac{885.6}{447.4}=1.87$$

The objective lens of confocal system has a larger numerical aperture, which directly leads to a higher fluorescence collection efficiency. While the diamond volume integrated in the optical fiber system is much larger, and the final fluorescence intensity detected by the optical fiber system is higher. It can be concluded that there is a good consistency between the fluorescence collection theoretical calculation and experiment results for both the optical fiber system and the confocal system. This model could be used to estimate the optical interrogation of NV diamond. However, when the diamond size is close to or smaller than the fiber core diameter, parts of the laser irradiate out of the fiber and do not pump the diamond. The utilization rate of the laser decreases, and the model needs to be modified with the laser utilization rate.

### 3.2. Sensitivity Analysis

Pulsed ODMR experiments are implemented by measuring the normalized change of fluorescence intensity as a function of the microwave frequency Ω. When an external magnetic field ***B*** is applied to NV centers, Zeeman frequency splitting ∆Ω between *m_s_* = ±1 sublevels is induced. Since the fluorescence intensity from *m_s_* = ±1 state NV centers is weaker than *m_s_* = 0 state, when the microwave frequency Ω resonates with a transition between the *m_s_* = 0 and *m_s_* = ±1 sublevels, the fluorescence intensity decreases. Four pairs of resonance peaks Ω*_i_*_±_ (i = 1, 2, 3, 4) would be generated corresponding to different projection (*B_i_*) of the external magnetic field on four NV axes directions [29], as demonstrated in Equation (5). *D* and *E* are the zero-magnetic-field splitting and lattice stress parameter, γ = 2.8 MHz/Gauss is the gyromagnetic ratio. The magnitude of ***B*** can be measured directly through Ω*_i_*_±_ according to Equation (6), in which ∆Ω*_i_ =* Ω*_i+_* − Ω*_i−_*.
(5)$$\Omega_{i\pm }=D\pm {\left(\gamma B_{i}\right)}^{2}+E^{2}$$
(6)$$\frac{4}{3}B^{2}=\frac{\sum {\Delta \Omega }_{i}{}^{2}-16E^{2}}{4\gamma^{2}}$$

The magnetometer sensitivity *η* is limited by photon shot-noise, which is estimated by Equation (7).
(7)$$\eta =\frac{4}{3\sqrt{3}}\cdot \frac{1}{\gamma }\cdot \frac{\Delta \nu }{C}\cdot \frac{1}{\sqrt{R}}$$
where ∆*ν* is the linewidth of the resonance peak in ODMR spectrum, *C* is the fluorescence intensity contrast, and *R* is the detected photon rate. The ODMR spectrum of the confocal system and fiber system were obtained with 20 Gauss DC magnetic field applied in the axial direction of the objective and fiber, which coincided with the <111> crystal direction of the diamond. As shown in Figure 5, the ODMR spectrum presented two pairs of resonance peaks, since the magnetic field along the <111> crystal direction has the same projection size on the other three NV axes. Lorentz fitting was performed on the resonance peaks in the box in Figure 5 to obtain the values of ∆*ν*_1_ and *C*_1_ for each system. For the confocal system, ∆*ν*_1_ = 8.8 MHz and *C*_1_
*=* 4.6% were obtained, while for the fiber system, ∆*ν*_2_ = 10.2 MHz and *C*_2_
*=* 3.2% were obtained. Wavelength of 700 nm was used to estimate the detected photon rate, *R*_1_ = 0.78 × 10^14^ Hz and *R*_2_ = 1.45 × 10^14^ Hz were obtained for confocal and fiber systems, respectively. Finally, the photon shot-noise limited sensitivity was 0.73 nT/Hz^1/2^ for all-fiber magnetometer and 0.59 nT/Hz^1/2^ for spatial optical confocal magnetometer.

Although, the all-fiber NV center magnetometer had higher fluorescence intensity. The all-fiber NV center magnetometer had a slightly lower sensitivity because of the wider linewidth and smaller contrast, compared with the confocal system. In the confocal system, laser was focused by the objective, resulting higher laser intensity distribution near the focal point where was the laser excited and fluorescence collected area of the diamond. The optical pumping rate of NV centers is proportional to the laser intensity. The ODMR spectra linewidth narrowing and contrast enhancement effect with increasing pumping rate has been discussed by Jensen et al. [30]. Compared with the fiber system, the linewidth of the confocal system deceased from 10.2 MHz to 8.8 MHz, and the contrast boosted from 3.2% to 4.6%, which agreed with Jensen, K. et al. ’s research result. Consequently, though a little sensitivity was sacrificed, the all-fiber magnetometer become robust and compact, could be applied for remote-magnetic measurement out of lab.

### 3.3. Vector Magnetic Field Measurement

There are challenges in defining the direction of the vector magnetic field in the lab frame. Firstly, the orientations of the four NV axes of the micro-diamond in lab frame are not easy to determine. Secondly, it is essential to distinguish these four NV axes and their corresponding resonance peaks and determine if *B_i_* is positive or negative [31].

Here, the 1 axis Helmholtz coil was introduced to solve the above-listed problems. The measurement steps of vector static magnetic field are illustrated in Figure 6. We set the angle between the auxiliary calibration magnetic field and the fiber to be 45° (this angle could be set to be any value; here we set it to 45° for the convenience of experiments and calculations). The C3v symmetry crystal lattice of diamond determines four possible orientations of NV center axes, as shown in Figure 7a. A Cartesian system of coordinates is built relative to fiber and the Helmholtz coil. One of the <111> crystal orientations in diamond, which is defined as axis 4, coincides with the axial direction of the optical fiber and is determined as *z*-axis. The *x*-, *y*-axes are introduced to set the auxiliary calibration magnetic field ***B****_a_*= (*B_0_*/2, *B_0_*/2, *B*_0_/√2) to facilitate testing and calibration. The projection of axes one, two, three and ***B****_a_* on the x-y plane is shown in Figure 7b. The axis one of NV centers is determined as its projection is the first axes that the *x*-axis encountered when it rotated counterclockwise in the *x-y* plane. We define the angle between axis one and the *x*-axis as *θ* (*θ*∈(0, 120°)) in *x-y* plane. We define ***B****_NV_*= (*B_1_, B_2_, B_3_, B_4_*) as the components in NV centers axes coordinate system, ***B****_lab_*= (*B_x_, B_y_, B_z_*) as the components in lab-based Cartesian system. The transformation between the lab-based Cartesian system and the NV centers axes coordinate system is described as Equation (8). ***K*** is the transformation matrix formed by the unit vectors of the four NV centers axes.
(8)$${B_{\mathrm{NV}}}^{T}=K{B_{\mathrm{lab}}}^{T},\;\left[\begin{array}{c} B_{1}\\ B_{2}\\ B_{3}\\ B_{4} \end{array}\right]=\left[\begin{array}{ccc} \frac{2\sqrt{2}}{2}\cos\theta & \frac{2\sqrt{2}}{2}\sin\theta & -\frac{1}{3}\\ \frac{2\sqrt{2}}{2}\cos\left(\theta +120^{\circ }\right) & \frac{2\sqrt{2}}{2}\sin\left(\theta +120^{\circ }\right) & -\frac{1}{3}\\ \frac{2\sqrt{2}}{2}\cos\left(\theta +240^{\circ }\right) & \frac{2\sqrt{2}}{2}\sin\left(\theta +240^{\circ }\right) & -\frac{1}{3}\\ 0 & 0 & 1 \end{array}\right]\left[\begin{array}{c} B_{x}\\ B_{y}\\ B_{z} \end{array}\right]$$

With an auxiliary calibration magnetic field ***B****_a_* generated by the Helmholtz coil with a 4A current, ODMR spectra were obtained, as Figure 8a shows. From the zero magnetic field measurements (shown as the inset in Figure 8a), *D* = 2871.5 MHz, *E* = 3.5 MHz were obtained. According to the result of the spectra fitting, we obtain differential resonance peaks ∆Ω*_ai_* of 216 MHz, 183 MHz, 82.5 MHz, 50.5 MHz, which are associated with absolute value of *B_ai_* 38.55 Gauss, 32.65 Gauss, 14.67 Gauss, and 8.91 Gauss, respectively. Using Equation (5), we could further calculate *B_a_* = 46.2 Gauss, ***B****_alab_*= (23.1, 23.1, 32.67) Gauss. Different arrangements of *B_ai_* were put into Equation (8), and there was only one solution *θ* = 78.69°, ***B****_NVa_*= (14.67, −38.55, −8.91, 32.65) Gauss. Therefore, four possible orientations of the NV center axes relative to the laboratory had been calibrated.

The ODMR spectrum of the static magnetic field to be measured ***B****_m_* was shown in Figure 8b. To distinguish the NV axes and their corresponding resonance peaks, the auxiliary calibration magnetic field ***B****_a_* was applied with ***B****_m_* at the same time, and the ODMR spectrum of the magnetic field ***B****_a_ + **B**_m_* was measured, as shown in Figure 8c. The resonance peaks in Figure 8c shifted compared to the spectrum in Figure 8b. The amount of shift for each peak is defined as *d*Ω*_i_* = ∆Ω*_ci_* − ∆Ω*_bi_*. We resolved the axes corresponding to each pair of resonance peaks through a numeric relationship |*d*Ω*_i_*| = ∆Ω*_ia_*. We can further determine if *B_mi_* is positive or negative through ∆Ω*_ia_* = ±*d*Ω*_i_*. Finally, we could obtain ***B****_NVm_ =* (1.19, 6.23, −15.61, 8.2) Gauss in the NV centers axis frame, corresponding to ***B****_labm_*= (−12.3, 6.7, 8.2) Gauss.

## 4. Conclusions

We demonstrate an all-fiber magnetometer with a sensitivity of 0.73 nT/Hz^1/2^, which is portable and can be quickly built outside of a laboratory. Due to the good optical anti-interference of the fiber system, this sensor system could work in more complex working conditions which promotes the practical use of NV center sensors. This all-fiber magnetometer is a suitable candidate for endoscopic measurement of remote inaccessible places. This schematic could also be expanded to thermometer and magnetic field imaging. An optical interrogation model of NV centers in diamond was established, which was essential to estimate the optical performance of the NV center system. A simplified rapid vector magnetic field measurement method is proposed for diamond particles with a one-axis Helmholtz coil, realizing μm-scale vector magnetic field detection.

## Figures and Tables

**Figure 1 nanomaterials-13-00949-f001:**
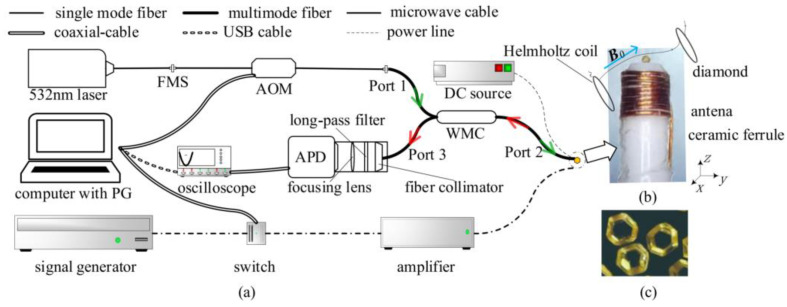
(**a**) Schematic of the all-fiber magnetometer setup. (**b**) A micro-diamond is attached on the tip of WMC’s Port 2 with an eight-turn copper coil twining around the fiber. A one-axis Helmholtz coil is settled near the micro-diamond to generate an auxiliary calibration magnetic field Ba. (**c**) The micro diamond with a diameter of 550 μm is observed under microscope.

**Figure 2 nanomaterials-13-00949-f002:**
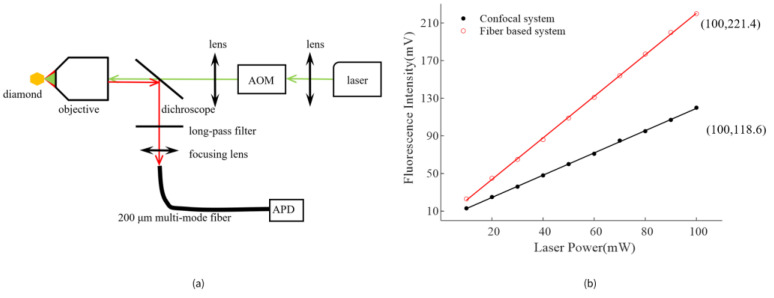
(**a**) Setup of the confocal system, a diameter of 200 μm multi-mode fiber was used as a pinhole. (**b**)The fluorescence intensity *I_0_* as function of laser source power in confocal system and fiber-based system. The same micro-diamond was used to reduce the influence of diamond morphology. The fluorescence intensity is described using APD output voltage value.

**Figure 3 nanomaterials-13-00949-f003:**
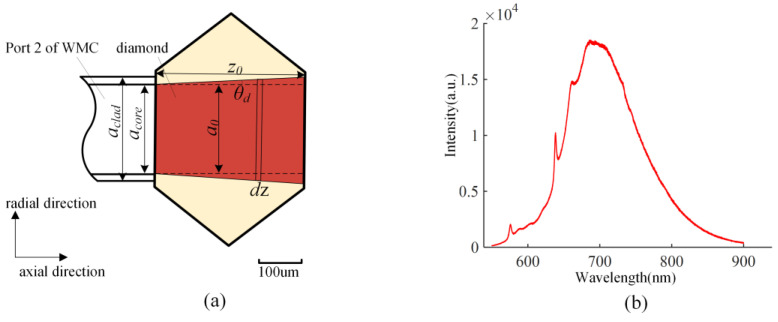
(**a**) The multi-mode fiber interrogation of micro-diamond, *a_core_* = 200 μm, diameter of the fiber core; *a_clad_* = 225 μm, diameter of the fiber cladding. The red area in the figure is the area where the diamond is excited. (**b**) Photoluminescence spectrum of NV centers.

**Figure 4 nanomaterials-13-00949-f004:**
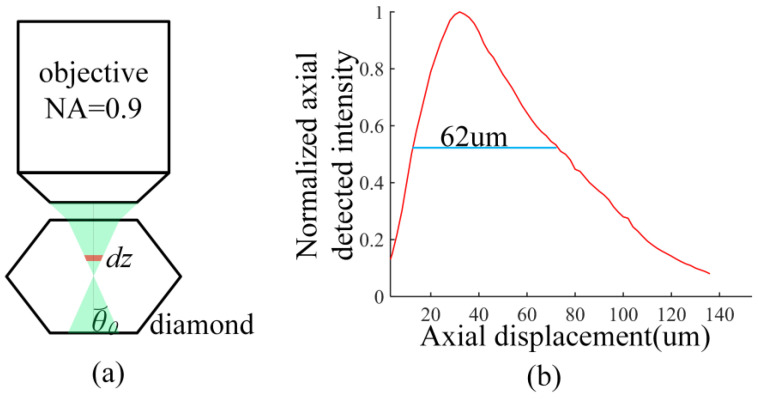
(**a**) The integration of objective lens (NA_2_ = 0.9, 100×) and micro-diamond in the confocal system. The diagram is not drawn to actual scale for clarity. (**b**) The detected axial optical sectioning property of the confocal system.

**Figure 5 nanomaterials-13-00949-f005:**
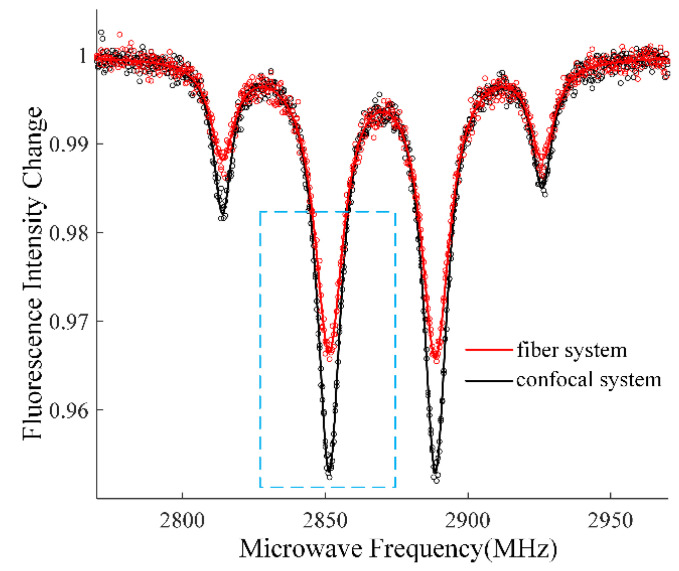
ODMR spectrum with 20 Gauss magnetic field.

**Figure 6 nanomaterials-13-00949-f006:**
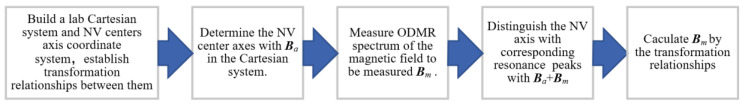
Measurement steps of vector static magnetic field.

**Figure 7 nanomaterials-13-00949-f007:**
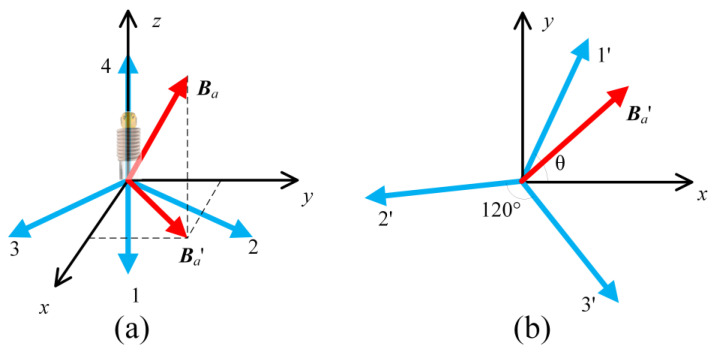
(**a**) Four possible orientations of NV center axes (blue arrow) with lab-based Cartesian system according to fiber and calibration magnetic fields (red arrow). (**b**) Axes 1′, 2′, 3′ are the projection of Axes 1, 2, 3 on the x-y plane, the angle between them is 120°.

**Figure 8 nanomaterials-13-00949-f008:**
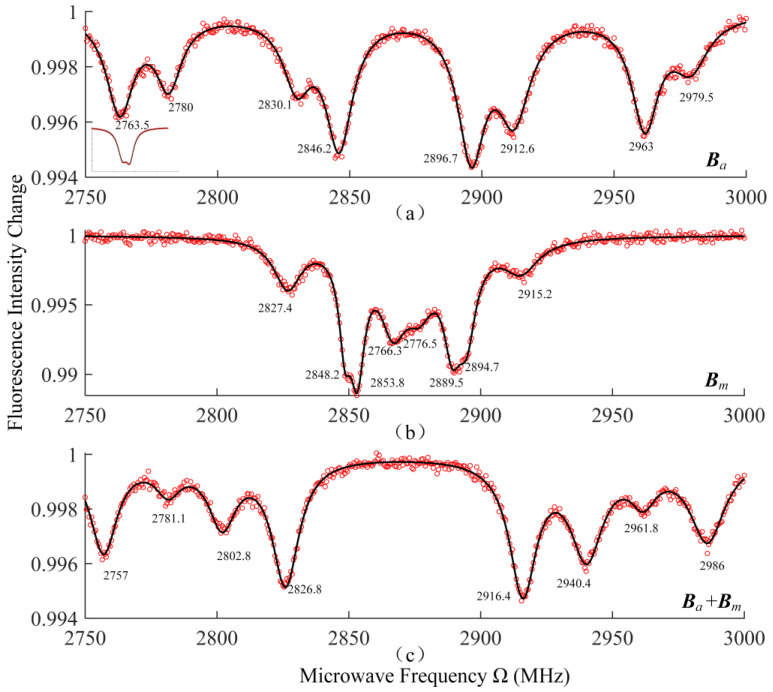
(**a**–**c**) are the ODMR spectra of calibration magnetic field ***B***_a_, measured magnetic field ***B****_m_*, and superimposed magnetic field ***B****_a_* + ***B**_m,_* respectively. The bottom left inset of (**a**) shows the ODMR spectrum at zero magnetic field. The data in the figure are the frequency of resonance peaks Ω*_ai_*, Ω*_bi_*, Ω*_ci_* (i = 1, 2, 3, 4 are the four orientations of NV center axes).

## Data Availability

Data underlying the results presented in this paper are not publicly available at this time but may be obtained from the authors upon reasonable request.

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
