# Peer review of "All Fiber Vector Magnetometer Based on Nitrogen-Vacancy Center"

_nanomaterials, 2023, doi:10.3390/nano13050949_

Round 1

Reviewer 1 Report

The article is well written and if there are any factual errors, I was not knowledgeable enough to find them.

I have only a few minor objections to the wording and to the presentation of information in the text or figures.

Line 32:  … have been widely used in temperature measurement of single living cells [7]

But: in my understanding, ref.7 does not focus on measurement of temperature of cells but their magnetic signatures. Perhaps this should be: “… have been widely used in room-temperature measurement of single living cells [7]”

Line 38-40: Fiber based NV center sensors … [11-15]. But: not all ref.11-15 deal with _fiber based_ sensing

Line 43: But they still remain several space light optical elements – Probably should be: : But there still remain

Line 56: 3-axis Helmholtz coil… – for the sake of smooth reading probably should be: In some published works by other groups a 3-axis Helmholtz coil…

Line 56: optical – optically

Line 69: magnetometer was shown – is shown

Figure 1: the signal connection between APD and oscilloscope is done in style assigned to USB cable. This should probably be rather coaxial cable(?). Also, colors used for the fiber mode designation are red and green which is rather unfortunate. In most articles dealing with NV diamond sensors, the red and green colors are used to show path of excitation (green) and response (red) light. I assume that single-mode/multi-mode fiber could be shown by (e.g.) solid/dashed line and the color should follow the wavelength (that is, if I understand correctly: AOM-WMC green, WMC-Port2 green+red, WMC-APD red).

Line 113: … a low insertion loss, typically less than 0.5dbm – it seems a little confusing to specify insertion loss in dBm rather than dB. It makes sense only in the connection with specifying incident power (which you mention in Materials and methods to be 100mW, if I understand correctly?)

Line 136: And NV centers in micro-diamond could be treated as an uniformly radiated – radiating(?)

Line 137: The emitted fluorescence intensity distribution(Ip) of NV centers is 1/4π. – Sorry, I do not understand. Please provide a little more context. Maybe I am missing something that is well known and self-evident fact in the field, but I lack that insight.

Line 154: Pfluo = … - in Eq.(2) symbol Pf is used rather than Pfluo. If you mean the same quantity as in (2), please use the same symbol, otherwise provide additional information.

Line 159: diviation – deviation

Line 231: the orientations of the four NV axis – axes (also in lines 244-246: singular axis, plural axes)

Line 244: The axes [axis] 1 of NV centers is determined as its projection is the first axes [axis] that the x-axis encountered when it rotated counterclockwise … we define the angle between axes [axis] 1 and the x-axis

Figure 6: in boxes 2, 3 and 4 a reference to Figure 4a, 4b and 4c is made. This cannot be correct. Probably should be reference to Figure 8a,b,c.

Line 260: ODMR spectra was obtained – were obtained (plural)

Line 260: . From the inset of zero magnetic field measurements – probably should be: From the zero magnetic field measurements (shown as the inset in Fig. 8a), D=…

Line 263-264: Guass (4x) – should be Gauss

Line 280 (Figure 8 label): (a)(b)(c) are the ODMR spectrum  -  spectra

Line 286: a sensitivity of 0.73nT/Hz1/2. – 1/2 should be in superscript (small typesetting error)

Author Response

Response to Reviewer 1 Comments

Point 1:  Line 32:… have been widely used in temperature measurement of single living cells [7]. But: in my understanding, ref.7 does not focus on measurement of temperature of cells but their magnetic signatures. Perhaps this should be: “… have been widely used in room-temperature measurement of single living cells [7]”

Response 1: We sincerely thank the reviewer for careful reading. As suggested by the reviewer, we have modified as your advice.

Point 2: Line 38-40: Fiber based NV center sensors … [11-15]. But: not all ref.11-15 deal with fiber based sensing

Response 2: We sincerely appreciate the valuable comments. We have checked the literature carefully. The ref.11-15 are not all fiber based sensing. I have replaced the original ref.11 and 12 with the following references.

  1. Lanin, A.A.; Fedotov, I.V.; Ermakova, Y.G.; Sidorov-Biryukov, D.A.; Fedotov, A.B.; Hemmer, P.; Belousov, V.V.; Zheltikov, A.M. Fiber-optic electron-spin-resonance thermometry of single laser-activated neurons. Opt. Lett. 2016, 41, 5563-5566, doi:10.1364/ol.41.005563.
  2. Guo, Z.; Chen, G.; Gu, B.; He, W.; Jiang, H.; Wang, H.; Du, G. Microwave Field Imaging of Stripline Chip Based on Nitrogen-Vacancy Center Ensembles in Diamond. Acta Electronica Sinica 2020, 48, 2258-2262.

Point 3: Line 43: But they still remain several space light optical elements – Probably should be: : But there still remain

Response 3: Thanks for your suggestion. This sentence has been modified to “But there still remain several space light optical elements”.

Point 4: Line 56: 3-axis Helmholtz coil… – for the sake of smooth reading probably should be: In some published works by other groups a 3-axis Helmholtz coil…

Response 4: We sincerely thank the reviewer for careful reading. As suggested by the reviewer, we have modified this sentence to “In some published works by other groups a 3-axis Helmholtz coil…”

Point5: Line 56: optical – optically

Response 5: We were really sorry for our careless mistakes. Thank you for your reminder. We have corrected the “optical” into “optically”.

Point 6: Line 69: magnetometer was shown – is shown

Response 6: We were really sorry for our careless mistakes. Thank you for your reminder.

Point 7: Figure 1: the signal connection between APD and oscilloscope is done in style assigned to USB cable. This should probably be rather coaxial cable(?). Also, colors used for the fiber mode designation are red and green which is rather unfortunate. In most articles dealing with NV diamond sensors, the red and green colors are used to show path of excitation (green) and response (red) light. I assume that single-mode/multi-mode fiber could be shown by (e.g.) solid/dashed line and the color should follow the wavelength (that is, if I understand correctly: AOM-WMC green, WMC-Port2 green+red, WMC-APD red).

Response 7: The the signal connection between APD and oscilloscope is done in style of coaxial cable. I have revised that. Thanks for your careful checks. We are sorry for our carelessness. We think this is an excellent suggestion about the colors used for the fiber mode designation. After the correction, I use lines with different thicknesses to represent the multimode fibers and multimode fibers. And the blue and red arrows were used to show path of the laser and fluorescence.

Point 8: Line 113… a low insertion loss, typically less than 0.5dbm – it seems a little confusing to specify insertion loss in dBm rather than dB. It makes sense only in the connection with specifying incident power (which you mention in Materials and methods to be 100mW, if I understand correctly?)

Response 8: This is a typo that should not have been there and I have corrected it to “dB”.

Point 9: Line 136 And NV centers in micro-diamond could be treated as an uniformly radiated – radiating(?)

Response 9: We feel sorry for our carelessness. In our resubmitted manuscript, the error has been revised to “radiating”. Thanks for your correction.

Point 10: The emitted fluorescence intensity distribution(Ip) of NV centers is 1/4π. – Sorry, I do not understand. Please provide a little more context. Maybe I am missing something that is well known and self-evident fact in the field, but I lack that insight.

Response 10: Because NV centers in micro-diamond could be treated as an uniformly radiating point light source. The energy radiated by each NV center is normalized to 1. The steradian of a sphere is 4π. Thus, the emitted fluorescence intensity distribution(Ip) of NV centers is 1/4π.

Point 11: Pfluo = … - in Eq.(2) symbol Pf is used rather than Pfluo. If you mean the same quantity as in (2), please use the same symbol, otherwise provide additional information.

Response 11: Thanks for your careful checks. We are sorry for our carelessness. Based on your comments, we have made the corrected “Pfluo” to “Pf”.

Point 12: Line 159: diviation – deviation

Response 12: We feel sorry for our carelessness. In our resubmitted manuscript, the typo is revised. Thanks for your correction.

Point 13: Line 231: the orientations of the four NV axis – axes (also in lines 244-246: singular axis, plural axes)

Response 13: We were really sorry for our careless mistakes. In our resubmitted manuscript, the errors have been revised. Thanks for your correction.

Point 14: Line 244: The axes [axis] 1 of NV centers is determined as its projection is the first axes [axis] that the x-axis encountered when it rotated counterclockwise … we define the angle between axes [axis] 1 and the x-axis

Response 14: We feel sorry for our carelessness. In our resubmitted manuscript, these errors have been revised. Thanks for your correction.

Point 15: Figure 6: in boxes 2, 3 and 4 a reference to Figure 4a, 4b and 4c is made. This cannot be correct. Probably should be reference to Figure 8a,b,c.

Response 15: Thanks for your careful check. We feel sorry for our carelessness. In our resubmitted manuscript, the mistake has been revised. Thanks for your correction.

Point 16: Line 260: ODMR spectra was obtained – were obtained (plural)

Response 16: We sincerely thank the reviewer for careful reading. As suggested by the reviewer, we have corrected the “was” into “were”.

Point 17: Line 260: From the inset of zero magnetic field measurements – probably should be: From the zero magnetic field measurements (shown as the inset in Fig. 8a), D=…

Response 17: We think this is an excellent suggestion. We have made change as your suggestion.

Point 18: Line 263-264: Guass (4x) – should be Gauss

Response 18: We were really sorry for our careless mistakes. In our resubmitted manuscript, the errors have been revised in the whole manuscript. Thanks for your correction.

Point 19: Line 280 (Figure 8 label): (a)(b)(c) are the ODMR spectrum  -  spectra

Response 19: We sincerely thank the reviewer for careful reading. As suggested by the reviewer, we have corrected the “spectrum” into “spectra”.

Point 20: Line 286: a sensitivity of 0.73nT/Hz1/2. – 1/2 should be in superscript (small typesetting error)

Response 20:We sincerely thank the reviewer for careful reading. As suggested by the reviewer, we have corrected the typesetting error.

Reviewer 2 Report

I believe that the manuscript is written at a good level and is of undoubted interest to the readers of the journal. I certainly recommend it for publication after the correction of minor remarks:

1.            All designations of physical units must be separated from numbers by spaces. Also spaces are required between references and text.

2.            Some sentences are unreadable or read ambiguously, for example “But they still remain several space light optical elements which are unstable and inflexible” (L43); “Optical interrogation analysis of micro-diamond in the confocal system and fiber system was investigated” - analysis was investigated? (L52), “optical (instead of “optically”) detected magnetic resonance” (L56), and so on.

3.            L72: “The WMC separated 532nm laser and 600-800nm fluorescence emitted from diamond effectively, replacing the traditional dichroic mirror” - According to the description of this device on the Thorlabs website, it separates incoming light from returning light, but does not have spectral selectivity. This sentence needs to be reformulated.

4.            L70: “a fiber connected acoustooptic modulator” - the purpose of this modulator requires clarification.

5.            L78: “The fiber collimator, long-pass filter, and focusing lens” - The term "long-wavelength filter" fits better.

6.            L99: “The micro-diamond was … encapsulated by a ceramic ferrule”. But from figure 1 it follows that the diamond is outside the ferrule! I would like to see a clearer drawing.

7.            L.114: “The laser power transmitted to the diamond reached 64.2mW when the laser source power was set to 100mW... In contrast to this, the laser transmission efficiency of traditional space light confocal system was about 62.3%, as shown in Figure 2(a)” - I don't see much contrast between 64.2% and 62.3%.

8.            L208: “20Guass”.

9.            L223: “The smaller optical pumping rate of 223 fiber system leads to the linewidth enlargement” – this statement need to be clarified, since usually light broadens the resonance. Note that Fig.5 shows neither broadening nor narrowing of resonances in the confocal scheme.

10.          L232: “it is essential to distinguish these four NV axis and their corresponding resonance peaks, and determine if Bi is positive or negative” – perhaps, the method first described in https://link.springer.com/article/10.1134/S1063785015040306.

Author Response

Response to Reviewer 2 Comments

Dear reviewer,

On behalf of my co-authors, we thank you very much for giving us an opportunity to revise our manuscript. We appreciate the reviewer very much for your positive and constructive comments and suggestions on our manuscript. Those comments are all valuable and very helpful for revising and improving our paper, as well as the important guiding significance to our researches. We have studied reviewer’s comments carefully and have made revision, which could be traced in the paper. We have tried our best to revise our manuscript according to the comments.

Best regards,

<Man Zhao> et al.

All the typesetting, grammar and spelling errors raised by the reviewers have been properly corrected in the manuscript. The problems that some sentences are unreadable or read ambiguously have also been revised as your suggestion. The main corrections in the paper and the responds to the reviewer’s comments are as follows:

Point 1:  All designations of physical units must be separated from numbers by spaces. Also spaces are required between references and text.

Response 1: Thanks for your careful checks. We are sorry for our carelessness. Based on your comments, we have made the corrections within the whole manuscript.

Point 2:  Some sentences are unreadable or read ambiguously, for example “But they still remain several space light optical elements which are unstable and inflexible” (L43); “Optical interrogation analysis of micro-diamond in the confocal system and fiber system was investigated” - analysis was investigated? (L52), “optical (instead of “optically”) detected magnetic resonance” (L56), and so on.

Response 2: Thanks for your careful check. We are sorry for our carelessness. Based on your comments, we have modified the whole manuscript to make these sentences clear and readable.

Point 3: “The WMC separated 532nm laser and 600-800nm fluorescence emitted from diamond effectively, replacing the traditional dichroic mirror” - According to the description of this device on the Thorlabs website, it separates incoming light from returning light, but does not have spectral selectivity. This sentence needs to be reformulated.

Response 3: Thanks for your suggestion. Here's the problem with misrepresentation. About 98% of the laser power from Port 1 was transmitted to pump the diamond which was glued at Port 2. About 92% of the fluorescence collected by Port 2 largely spread to Port 3. And the light transmitted from Port 1 to 3 and Port 2 to 1 was isolated. Finally, the laser and fluorescence is separated by the WMC effectively. It is the ability of separating incoming light from returning light that separates the laser and fluorescence, but not wavelength selectivity. I have reformulated the sentences in the manuscript.

Point 4: L70: “a fiber connected acoustooptic modulator” - the purpose of this modulator requires clarification.

Response 4: The fiber connected acoustooptic modulator was used for fast switch of laser to realize pulse ODMR measurement. We have modified in the manuscript.

Point 5: L78: “The fiber collimator, long-pass filter, and focusing lens” - The term "long-wavelength filter" fits better.

Response 5: Thanks for your suggestion. I changed “long-pass filter” to "long-wavelength filter".

Point 6: L99: “The micro-diamond was … encapsulated by a ceramic ferrule”. But from figure 1 it follows that the diamond is outside the ferrule! I would like to see a clearer drawing.

Response 6: Here is my misleading expression. To improve the mechanical strength of the probe, the fiber end near the micro diamond was encapsulated by a ceramic ferrule. And the copper coil antenna was wound around the ceramic ferrule. But the diamond is exactly outside the ferrule. We have reformulated the sentences in the manuscript.

Point 7: L.114: “The laser power transmitted to the diamond reached 64.2mW when the laser source power was set to 100mW... In contrast to this, the laser transmission efficiency of traditional space light confocal system was about 62.3%, as shown in Figure 2(a)” - I don't see much contrast between 64.2% and 62.3%.

Response 7: Thanks for your careful check. This expression is misleading. There is not much difference between the laser transmission efficiency of traditional space light confocal system and the fiber system. So, we deleted the expression of “In contrast to this”.

Point 8: L208: “20Guass”.

Response 8: We feel sorry for our carelessness. In our resubmitted manuscript, the typo is revised. Thanks for your correction.

Response 9: L223: “The smaller optical pumping rate of 223 fiber system leads to the linewidth enlargement” – this statement need to be clarified, since usually light broadens the resonance. Note that Fig.5 shows neither broadening nor narrowing of resonances in the confocal scheme.

Response 9: Here I quote the conclusion of Reference 30. [Light narrowing of magnetic resonances in ensembles of nitrogen-vacancy centers in diamond]. This paper demonstrated that the increased pumping rate of NV centers would cause linewidth narrowing and contrast enhancement in theory and experiment. And the optical pumping rate of NV centers is proportional to the laser intensity. In this paper, the linewidth ∆ν and contrast C of the ODMR resonance are depicted on the following form:

In the formula, p is the laser intensity, and cP is the laser pumping rate. In theory, this explains why the increased intensity of the laser would cause linewidth narrowing and contrast enhancement. In addition, The linewidth ∆ν and contrast C as function of laser power was measured with different Rabi frequency (as shown in following figures), which agreed with the above theory relationship.

In my paper, I used his conclusion directly to explain the experimental result. By Lorentz fitting the resonance peaks in the box in Figure 5, Δ?1=8.8 MHz and C1=4.6% in the confocal system, Δ?2=10.2 MHz and C2=3.2% in the fiber system were obtained. The linewidth decreased from 10.2 MHz to 8.8 MHz, which is not particularly significant, so it is not particularly easy to observe the narrowing effect in the Figure 5. The smaller optical pumping rate of fiber system leads to the linewidth enlargement and contrast decline. This experimental result is consistent with the conclusions in the ref.30.

Point 10: L232: “it is essential to distinguish these four NV axis and their corresponding resonance peaks, and determine if Bi is positive or negative” – perhaps, the method first described in https://link.springer.com/article/10.1134/S1063785015040306.

Response 10: We sincerely appreciated the valuable comments and read the paper very carefully. This paper used a weak modulating magnetic field to distinguish NV center and their corresponding resonance peaks with different magnetic field direction. It is very innovative and instructive. Thus, I added this paper as the reference 31 for my manuscript.

31.Vershovskii, A.K.; Dmitriev, A.K. Micro-Scale Three-Component Quantum Magnetometer Based on Nitrogen-Vacancy Color Centers in Diamond Crystal. Technical Physics Letters 2015, 41, 393-396, doi:10.1134/s1063785015040306.

Reviewer 3 Report

The article proposes the design and demonsatrates the performance of all-fiber magnetometer based on nitrogen-vacancy (NV) centers in diamond. Unlike existing NV-center magnitometers, utilizing several discrete optical elements to excite and collect the photoluminescence, the proposed device uses the micro-diamond glued directly to the tip of the optical fiber, which allowed, with only a slight decrease of sensitivity, to realize extremely compact and robust magnitometer (sensing part of which fits on the tip of the optical fiber, no "trimming" parts required). Such magnitometers may have many possible practical application. The arcicle is generally well written, presents a detailed analisys of NV-centers magnitometers, and clearly deserves publishing.

But althoug the language of the article is generally clear and understandable, there are words and phrases that sound peculiar to me, so I think the article still needs a touch of language editing. Examples that I've noticed:

L 43: "But they still remain several space light optical elements which are unstable and inflexible" - "still have several optical elements that needs to be precisely spaced, which makes the system unstable and inflexible"?

L 51: "can be quickly set up outdoors" - "...outside the lab [envinroment]"?

L 61: "combining the morphology of micro-diamond" - combining with what? "Considering", "taking into account"? 

L 84: "lastly" - "finally"?

L 193: "analyzation" - "analysis"?

L 269: "spectrum of the to be measured static magnetic field" - "spectrum of the static magnetic field to be measured"?

Author Response

Response to Reviewer 3 Comments

Dear reviewer,

On behalf of my co-authors, we thank you very much for giving us an opportunity to revise our manuscript. We appreciate the reviewer very much for your positive and constructive comments and suggestions on our manuscript. Those comments are all valuable and very helpful for revising and improving our paper, as well as the important guiding significance to our researches. We have studied reviewer’s comments carefully and have made revision, which could be traced in the paper. We have tried our best to revise our manuscript according to the comments.

Best regards,

<Man Zhao> et al.

All the typesetting, grammar and spelling errors raised by the reviewers have been properly corrected in the manuscript. The problems that some sentences are unreadable or read ambiguously have also been revised as your suggestion. The main corrections in the paper and the responds to the reviewer’s comments are as follows:

Point 1: L 43: "But they still remain several space light optical elements which are unstable and inflexible" - "still have several optical elements that needs to be precisely spaced, which makes the system unstable and inflexible"?

Response 1: We tried our best to improve the manuscript and made some changes to the manuscript. This sentence was modified to “But there still remain several space light optical elements that needs to be precisely spaced, which makes the system unstable and inflexible”.

Point 2: L 51: "can be quickly set up outdoors" - "...outside the lab [envinroment]"?

Response 2: Thanks for your suggestion. We have tried our best to polish the language in the revised manuscript. As your suggestion, this sentence was modified to “can be quickly set up outside the lab”.

Point 3: L 61: "combining the morphology of micro-diamond" - combining with what? "Considering", "taking into account"? 

Response 3: Thanks for your careful checks. This sentence was modified to “We also proposed a simplified and rapid approach for vector magnetic field measurement, considering the relationship between the morphology of micro-diamond and NV centers orientations.”

Point 4: L 84: "lastly" - "finally"?

Response 4: As suggested by the reviewer, we corrected "lastly" to "finally".

Point 5: L 193: "analyzation" - "analysis"?

Response 5:As suggested by the reviewer, we corrected "analyzation" to "analysis".

Point 6: L 269: "spectrum of the to be measured static magnetic field" - "spectrum of the static magnetic field to be measured"?

Response 6: Thanks for your careful check. As your suggestion, this sentence was modified to “spectrum of the static magnetic field to be measured"
